# Evaluation and Optimization of the Anti-Melanogenic Activity of 1-(2-Cyclohexylmethoxy-6-hydroxy-phenyl)-3-(4-hydroxymethyl-phenyl)-propenone Derivatives

**DOI:** 10.3390/molecules24071372

**Published:** 2019-04-08

**Authors:** Byung-Hak Kim, Soo-Nam Hong, Sang-Kyu Ye, Jung-Youl Park

**Affiliations:** 1Department of Pharmacology and Biomedical Sciences, Seoul National University College of Medicine, 103 Daehak-ro, Jongno-gu, Seoul 03080, Korea; protein0826@snu.ac.kr; 2Biomedical Science Project (BK21^PLUS^), Seoul National University College of Medicine, 103 Daehak-ro, Jongno-gu, Seoul 03080, Korea; 3Department of Beauty Care, Bucheon University, 25 Sinheung-ro 56 beon-gil, Bucheon-si, Gyeonggi-do 14632, Korea; sn1228@naver.com; 4Ischemic/Hypoxic Disease Institute, Seoul National University College of Medicine, 103 Daehak-ro, Jongno-gu, Seoul 03080, Korea; 5Neuro-Immune Information Storage Network Research Center, Seoul National University College of Medicine, 103 Daehak-ro, Jongno-gu, Seoul 03080, Korea; 6Department of Applied Chemistry, Daejeon University, 62 Daehak-ro, Dong-gu, Daejeon 34520, Korea

**Keywords:** alpha-melanocyte stimulating hormone (α-MSH), chalcone, melanin biosynthesis, (*E*)-*N*-(4-(3-(2-(cyclohexylmethoxy)phenyl)-3-oxoprop-1-en-1-yl)phenyl)acetamide (chalcone 21-21), tyrosinase

## Abstract

The chemical modification and optimization of biologically active compounds are essential steps in the identification of promising lead compounds for drug development. We previously reported the anti-melanogenic activity of 1-(2-cyclohexylmethoxy-6-hydroxy-phenyl)-3-(4-hydroxymethyl-phenyl)-propenone (chalcone 21). In this study, we synthesized 21 derivatives of chalcone 21 and evaluated their anti-melanogenic activity in α-MSH-induced B16F10 cells. (*E*)-*N*-(4-(3-(2-(Cyclohexylmethoxy)phenyl)-3-oxoprop-1-en-1-yl)phenyl)acetamide (chalcone 21-21) exhibited the strongest inhibition of cellular melanin production, with an IC_50_ value of 0.54 μM. It was more potent than chalcone 21 and the known anti-melanogenic agents kojic acid and arbutin, whose IC_50_ values were 4.9, 38.5, and 148.4 μM, respectively. Chalcone 21-21 decreased the expression and activity of tyrosinase. It also decreased the expression of TRP1, TRP2 and MITF, the phosphorylation of CREB and ERK1/2, and the transcriptional activity of MITF and CRE. Our results demonstrate that chalcone-21-21 is an effective lead compound with anti-melanogenic activity.

## 1. Introduction

Melanin is a group of dark polymer pigments synthesized in melanocytes. It is found in almost all animals, and is responsible for the various pigmentation of the skin, hair, and eyes. Several beneficial functions of melanin have been identified, including the protection of skin cancer development by protecting cells from damage due to sunlight, hormonal factors, cytokines, toxic drugs, and other chemicals [1,2]. However, abnormal loss or accumulation of melanin results in dermatological disorders such as vitiligo, freckles, melasma, chloasma, ephelides, senile lentigines, and melanoderma. In addition, melanin can induce skin inflammatory responses such as eczema, allergic contact dermatitis, and irritant contact dermatitis [3,4,5]. These disorders can lead to serious and emotionally distressing problems, and development of anti-melanogenic agents is necessary to address them. Melanin biosynthesis is mediated by a series of intricate signaling pathways initiated by many intrinsic and extrinsic stimuli, including ultraviolet (UV) radiation, nitric oxide (NO), stem cell factor (SCF), and hormones such as alpha-melanocyte stimulating hormone (α-MSH), epinephrine, and adrenocorticotropic hormone (ACTH). These stimuli activate many signaling pathways, such as those involving inositol trisphosphate/diacylglycerol (IP3/DAG)/protein kinase C (PKC), guanylate cyclase/cyclic guanosine monophosphate (GC/cGMP), c-Kit, and adenylate cyclase/cyclic adenosine monophosphate (AC/cAMP) [4,6,7].

Chalcones are aromatic ketones consisting of two aromatic rings linked to a three-carbon α,β-unsaturated carbonyl system as a central core. They are naturally occurring in plants that have been used in traditional folk medicine. Chalcones possess many biological effects, including anti-inflammatory, antimicrobial, antifungal, antioxidative, cytotoxic, antiproliferative, antitumor, and chemopreventive effects. In addition, chalcones are considered to be intermediates in the biosynthesis of flavonoids and isoflavonoids, abundant substances in edible plants that show a wide spectrum of biological activities [8,9,10,11,12]. We reported previously that the chalcone derivative 1-(2-cyclohexylmethoxy-6-hydroxy-phenyl)-3-(4-hydroxymethyl-phenyl)-propenone (chalcone 21) has anti-melanogenic activity [13]. In an attempt to increase its anti-melanogenic activity, we synthesized 21 derivatives via the aldol condensation of 2- and/or 6-substituted acetophenones and appropriate benzaldehydes in the presence of potassium hydroxide as a catalyst in ethanol. The largest difference with the derivatives previously reported is the presence of an hydroxyl group in the R_2_ position. Among the derivatives, we identified (*E*)-*N*-(4-(3-(2-(cyclohexylmethoxy)phenyl)-3-oxoprop-1-en-1-yl)phenyl)acetamide (chalcone 21-21) as exhibiting the strongest anti-melanogenic activity (approximately 10-fold more effective than chalcone 21).

## 2. Results and Discussion

### 2.1. Chemistry

We previously reported the anti-melanogenic activity of chalcone 21 [13]. To obtain more potent anti-melanogenic agents, we synthesized a series of chalcone derivatives by aldol condensation reactions (Scheme 1) [8,14]. Detailed synthetic procedures and the ^1^H-NMR and ^13^C-NMR spectroscopic data are shown in the Materials and Methods.

### 2.2. Biological Evaluation

#### 2.2.1. Cellular Melanin Production

(2*E*)-1,3-Diphenyl-propene-1-one forms the core structure of chalcones, consisting of two aromatic rings linked to a three-carbon α,β-unsaturated carbonyl (Figure 1A). To determine the anti-melanogenic activity of the 21 chalcone derivatives (21-1 through 21-21), we measured the inhibition of cellular melanin production in α-MSH-stimulated mouse melanoma B16F10 cells. α-MSH is the main intrinsic stimulator of melanogenesis, via a melanocortin receptor type 1 (MC1R)-mediated cAMP/protein kinase A (PKA) pathway [15,16]. Cells were incubated for 72 h with vehicle alone or with various concentrations of each compound in the presence of α-MSH, and melanin content was determined by measuring the absorbance at 405 nm of the cell-free culture medium. The cellular melanin levels were 34.4 ± 1.8 μg/mL in vehicle-treated resting cells, and were dramatically increased to 135.2 ± 2.5 μg/mL by α-MSH stimulation. The degree of inhibiton of the melanin production varied among the tested compound. For each, the percent inhibition of melanin production, the half maximal inhibitory concentration (IC_50_), and the cell viability of cultures exposed at 3 μM are shown in Table 1.

We observed that chalcones 21-5, 21-13, and 21-21 exhibited strong anti-melanogenic activity, with IC_50_ values of 0.88, 1.6, and 0.54 μM, respectively. Weak cytotoxic activity was associated with those compounds containing an acetamido or carboxyl group at the R_3_ position. Chalcones 2-2, 2-7, 2-9, and 2-11 exhibited moderate anti-melanogenic activity, with IC_50_ values of 1.2, 2.3, 2.6, and 2.0 μM, respectively; weak to moderate cytotoxic activity was observed for these forms containing a chloride atom at the R_3_ position. Other chalcone compounds with ethyl or methoxy group at the R_3_ position exhibited very weak to weak anti-melanogenic activity, and weak to moderate cytotoxic activity. The strong, moderate, and weak anti-melanogenic and cytotoxic activities were divided into over 75%, 50~75%, and less than 50% at 3 μM of each compound. Based on the structure-activity relationship (SAR), moderate to strong anti-melanogenic activity was observed when benzyloxy, 4-trifluoromethoxybenzyloxy or cyclohexylmethoxy group was located at the R_1_ position and chloride or acetamido group was located at the R_3_ position. However, hydrogen or hydroxyl group at the R_2_ position had little affected. In fact, benzyloxy, 4-trifluoromethoxybenzyloxy and chloride groups exhibited moderate cytotoxic activity, while cyclohexylmethoxy and acetamido groups exhibited weak cytotoxic activity. Overall, the anti-melanogenic activity of chalcone derivatives is the strongest when cyclohexylmethoxy and acetamido groups are located at the R_1_ and R_3_ positions. These data suggest the significance of acetamido or carboxyl groups at the R_3_ position to anti-melanogenic activity in the derivatives of chalcone 21. In particular, chalcone 21-21 exhibited the strongest inhibition of melanin production, and was approximately 10-fold more effective than chalcone 21 (Figure 1B and Table 1). These results are preliminary data at the simple screening step and require further analysis to determine the action mechanisms against anti-melanogenic activity.

#### 2.2.2. Anti-Melanogenic Activity of Chalcone 21-21

Because chalcone 21-21 exhibited the greatest inhibition of melanin production in α-MSH-stimulated B16F10 cells, we investigated the mechanisms underlying its anti-melanogenic effects. First, we determined the cytotoxic activity of chalcone 21-21. It exhibited weak to no cytotoxic activity in vitro at concentrations of up to 10 μM, in cells cultured for 24–72 h; above 10 μM, cytotoxic activity increased in concentration- and time-dependent manners. Treatment with 10 μM for 72 h showed weak to moderate cytotoxic activity, but the cytotoxic activity of chalcone 21-21 was not significantly different at concentrations of up to 100 μM in the absence or presence of α-MSH (Figure 2A,B). Therefore, chalcone 21-21 was used at concentrations of up to 3 μM in subsequent in vitro experiments to exclude the effects of cytotoxicity.

To determine the effect of chalcone 21-21 on melanin production, cellular melanin release was measured in B16F10 cells, followed by incubation for 72 h with various concentrations of chalcone 21-21 in the absence or presence of α-MSH. The released melanin levels were 31.3 ± 2.3 μg/mL in vehicle-treated resting cells and 139.8 ± 5.5 μg/mL in α-MSH-stimulated cells. Treatment with chalcone 21-21 decreased melanin release in a concentration-dependent manner, corresponding to 16.9 ± 5.5% inhibition at 0.1 μM, 38.4 ± 5.2% inhibition at 0.3 μM, 72.2 ± 5.2% inhibition at 1 μM, and 92.0 ± 4.2% inhibition at 3 μM, for an IC_50_ value of 0.54 μM (Figure 2C). Kojic acid and arbutin, well-known anti-melanogenic agents that act by inhibiting tyrosinase activity [17,18] were used as positive controls. Both compounds also reduced melanin production in a concentration-dependent manners with IC_50_ values of 38.5 μM and 148.4 μM, respectively (Figure 2D). These results indicate that chalcone 21-21 has significant potential anti-melanogenic activity.

#### 2.2.3. Inhibition of Tyrosinase Activity

Next, we evaluated whether chalcone 21-21 could inhibit tyrosinase activity. Tyrosinase is a copper-dependent enzyme that catalyzes the conversion of l-tyrosine to l-3,4-dihydroxyphenylalanine (l-DOPA), the rate-limiting step in melanin biosynthesis [19,20]. To determine the mRNA and protein levels of tyrosinase, B16F10 cells were incubated for 72 h with various concentrations of chalcone 21-21 in the absence or presence of α-MSH. Both the mRNA and protein levels of tyrosinase were just detectable in vehicle-treated resting cells, and the levels were markedly up-regulated by α-MSH stimulation. These up-regulated expression levels decreased by treatment with chalcone 21-21 in a concentration-dependent manner (Figure 3A).

Tyrosinase activity was further determined in cell lysates of B16F10 cells that were incubated for 72 h with various concentrations of chalcone 21-21 in the absence or presence of α-MSH. To perform zymography, lysate proteins were separated by native SDS-PAGE, and the gels were incubated with l-DOPA for the colorimetric detection of l-dopaquinone, which is the tyrosinase-catalyzed metabolite of l-DOPA and a precursor of melanin [21,22]. Tyrosinase activity was very weak in vehicle-treated resting cells, but was dramatically increased by α-MSH stimulation. Treatment with chalcone 21-21 decreased tyrosinase activity in a concentration-dependent manner (Figure 3B).

We also evaluated tyrosinase promoter activity to determine whether the expression of tyrosinase was regulated at the transcriptional level. B16F10 cells were transiently transfected with a tyrosinase reporter plasmid construct containing the human tyrosinase promoter together with the pRL-TK *Renilla* control vector. After stimulation with α-MSH, luciferase activity increased 7.4-fold over resting cells. Treatment with chalcone 21-21 decreased α-MSH-induced tyrosinase promoter activity in a concentration-dependent manner, corresponding to 14.5 ± 5.2% inhibition at 0.1 μM, 36.4 ± 4.4% inhibition at 0.3 μM, 70.7 ± 3.3% inhibition at 1 μM, and 94.5 ± 1.2% inhibition at 3 μM (Figure 3C). These results suggest that chalcone 21-21 has potential for development of skin-lightening agent.

#### 2.2.4. Inhibition of Melanogenesis-Related Proteins

Melanin biosynthesis occurs in melanosomes, membrane-bound organelles within melanocytes. In mammals, melanin biosynthesis is a tightly regulated complex network consisting of multiple important factors including tyrosinase-related protein (TRP)1 and TRP2, which share numerous structural similarities due to amino acid sequences that are 40% homologous to that of tyrosinase [1,20]. Therefore, we investigated whether chalcone 21-21 could affect the protein levels of melanogenesis-related TRP1 and TRP2 by Western blot analysis using whole-cell lysates from α-MSH-stimulated B16F10 cells. Both protein levels were hardly detectable in vehicle-treated resting cells, but their expression levels were dramatically increased by stimulation with α-MSH. Chalcone 21-21 treatment resulted in decreasing the α-MSH-induced protein synthesis of TRP1 and TRP2 in a concentration-dependent manner (Figure 4A).

Microphthalmia-associated transcription factor (MITF) is the master transcription factor in melanin biosynthesis, initiating the transcription of genes encoding tyrosinase, TRP1, and TRP2 [23]. The mRNA and protein levels of MITF were detactable in vehicle-treated resting B16F10 cells, and were dramatically increased by stimulation with α-MSH. Treatment with chalcone 21-21 decreased expression at both levels in a concentration-dependent manner (Figure 4B). We further determined the transcriptional regulation of MITF using a reporter system. A luciferase expression plasmid under the control of the human MITF promoter was transiently transfected together with the pRL-TK *Renilla* control vector into B16F10 cells. Stimulation with α-MSH increased luciferase activity 4.8-fold compared to the vehicle-treated resting cells. Treatment with chalcone 21-21 reduced α-MSH-induced MITF luciferase activity in a concentration-dependent manner, corresponding to 13.2 ± 3.9% inhibition at 0.1 μM, 36.2 ± 3.3% inhibition at 0.3 μM, 67.4 ± 3.6% inhibition at 1 μM, and 89.5 ± 3.3% inhibition at 3 μM (Figure 4C). These results indicate that chalcone 21-21 down-regulates multiple melanogenesis-related proteins including tyrosinase, TRP1, TRP2, and MITF, specifically via the transcriptional regulation of MITF.

#### 2.2.5. Inhibition of ERK and CREB Signaling

Canonical α-MSH-induced melanogenesis signaling is mediated by MC1R, which activates cAMP/PKA, resulting in the phosphorylation of the cAMP response element-binding protein (CREB) and regulation of MITF. CREB can act as a transcription factor to induce several genes including MITF, and elevated MITF activity is reinforced and regulated by extracellular signal-regulated kinase (ERK) activation. In addition, ERK2 can activate MITF by serine phosphorylation at residue 73 [1,6,20,23]. This indicates that the signaling pathways for melanin biosynthesis involve cross-talk, and may be closely linked. Therefore, we determined whether chalcone 21-21 could affect ERK1/2 and/or CREB. Phosphorylation of ERK1/2 and CREB was barely detectable in vehicle-treated resting B16F10 cells, but was dramatically increased by α-MSH stimulation. The levels of phosphorylated ERK1/2 and CREB decreased by treatment with chalcone 21-21 in a concentration-dependent manner (Figure 5A).

We also determined the transcriptional activity of CREB by measuring CRE-luciferase activity. B16F10 cells were transiently transfected with a CRE reporter plasmid construct containing the human CRE promoter together with the pRL-TK *Renilla* control vector. α-MSH stimulation increased luciferase activity by 6.1-fold over that in the vehicle-treated resting cells. α-MSH-induced CRE-luciferase activity decreased by chalcone 21-21 treatment in a concentration-dependent manner, corresponding to 14.0 ± 1.9% inhibition at 0.1 μM, 36.3 ± 3.3% inhibition at 0.3 μM, 69.5 ± 3.4% inhibition at 1 μM, and 92.6 ± 0.4% inhibition at 3 μM (Figure 5B). These results suggest that inhibition of ERK1/2 and CREB activities is an important target of chalcone 21-21 for anti-melanogenic activity.

## 3. Materials and Methods

### 3.1. Chemical Synthesis

To synthesize the derivatives of chalcone 21, anhydrous ethanol was added 2- and/or 6-substituted acetophenones, followed by KOH as a base, together with the appropriate benzaldehydes, at room temperature. The mixture was refluxed for 2~10 h, and then evaporated to remove the ethanol solvent. Methylene chloride was added to the concentrated solution, which was neutralized with HCl. The methylene chloride extract layer was dried with anhydrous MgSO_4_ and evaporated under reduced pressure. Reaction progress was monitored using an analytical thin layer chromatography (TLC) on pre-coated Merck silica-gel Kiesegel 60 F254 plates and the spots were detected under UV light (254 nm). Flash chromatography was conducted using a silica gel 230–400 mesh. Flash chromatography was performed, and 21 compounds were obtained from the liquid products (Table 1). Proton nuclear magnetic resonance (^1^H-NMR) and carbon-13 NMR (^13^C-NMR) spectra were recorded at 400 (^1^H-NMR) and 100 (^13^C-NMR) MHz on a Bruker ARX 400 (Bruker, Billerica, MA, USA) spectrometer and mass spectrometry (MS) data were collected using an AB SCIEX API 3200^TM^ (SCIEX, Ontario, Canada) LC-MS/MS system. The chemical shifts were reported downfield in ppm (δ) relative to internal tetramethylsilane (TMS), and coupling constants were reported in Hertz (Hz). All chemicals and solvents were purchased from Sigma–Aldrich (St. Louis, MO, USA).

#### 3.1.1. (*E*)-3-(4-Ethylphenyl)-1-(2-hydroxy-6-((4-(trifluoromethoxy)benzyl)oxy)phenyl)prop-2-en-1-one (21-1)

^1^H-NMR (400 MHz, CDCl_3_): δ 13.3 (s, 1H), 7.77 (d, *J* = 2.8 Hz, 1H), 7.50 (d, *J* = 8.8 Hz, 1H), 7.46 (d, *J* = 8.8 Hz, 1H), 7.45–7.35 (m, 3H), 7.20–7.15 (m, 3H), 6.66 (dd, *J* = 8.4, 24.4 Hz, 1H), 6.47 (dd, *J* = 8.0, 33.6 Hz, 1H), 5.13 (s, 2H), 2.36-2.29 (m, 3H), 1.24 (t, *J* = 7.6 Hz, 3H); ^13^C-NMR (100 MHz, DMSO-*d*_6_): d 194.6, 158.8, 157.9, 148.3, 147.2, 144.1, 136.6, 133.0, 132.4, 130.0, 129.0, 128.8, 128.0, 121.8, 121.3, 119.2, 115.9, 110.0, 104.0, 69.4, 28.5, 15.6; LC-MS (ESI) 443.2 (M + H).

#### 3.1.2. (*E*)-3-(4-Chlorophenyl)-1-(2-hydroxy-6-((4-(trifluoromethoxy)benzyl)oxy)phenyl)prop-2-en-1-one (21-2)

^1^H-NMR (400 MHz, CDCl_3_): δ 13.3 (s, 1H), 7.82–7.63 (m, 2H), 7.49 (d, *J* = 8.6 Hz, 2H), 7.40 (t, *J* = 8.3 Hz, 1H), 7.25–7.15 (m, 4H), 7.10 (d, *J* = 8.5 Hz, 2H), 6.68 (d, *J* = 8.4 Hz, 1H), 6.52 (d, *J* = 8.1 Hz, 1H), 5.13 (s, 2H); ^13^C-NMR (100 MHz, DMSO-*d*_6_): d 193.2. 158.6, 157.7, 146.5, 145.1, 136.9, 133.5, 133.3, 129.7, 129.3, 128.7, 128.0, 128.7, 114.5, 114.0, 110.5, 70.8; LC-MS (ESI) 489.1 (M + H).

#### 3.1.3. (*E*)-1-(2-Hydroxy-6-((4-(trifluoromethoxy)benzyl)oxy)phenyl)-3-(4-methoxyphenyl)prop-2-en-1-one (21-3)

^1^H-NMR (400 MHz, CDCl_3_): δ 13.5 (s, 1H), 7.82–7.60 (m, 2H), 7.52 (d, *J* = 8.4 Hz, 2H), 7.36 (t, *J* = 8.4 Hz, 1H), 7.26–7.10 (m, 4H), 6.76 (d, *J* = 8.8 Hz, 2H), 6.66 (d, *J* = 8.4 Hz, 1H), 6.51 (d, *J* = 8.4 Hz, 1H), 5.13 (s, 2H), 3.83 (s, 3H); ^13^C-NMR (100 MHz, DMSO-*d*_6_): d 194.4, 158.6, 157.7, 148.1, 146.9, 144.1, 136.4, 132.8, 132.2, 130.0, 129.0, 128.6, 128.0, 121.5, 121.3, 119.1, 115.7, 110.0, 104.0, 69.8, 55.8; LC-MS (ESI) 445.1 (M + H).

#### 3.1.4. (*E*)-4-(3-(2-Hydroxy-6-((4-(trifluoromethoxy)benzyl)oxy)phenyl)-3-oxoprop-1-en-1-yl)benzoic acid (21-4)

^1^H-NMR (400 MHz, CDCl_3_): δ 13.2 (s, 1H), 7.96 (d, *J* = 7.9 Hz, 2H), 7.84 (d, *J* = 15.8 Hz, 1H), 7.71 (d, *J* = 15.6 Hz, 1H), 7.54–7.38 (m, 3H), 7.34–7.13 (m, 5H), 6.68 (d, *J* = 8.6 Hz, 1H), 6.54 (d, *J* = 8.4 Hz, 1H), 5.14 (s, 2H); ^13^C-NMR (100 MHz, DMSO-*d*_6_): d 194.6, 158.3, 157.6, 148.3, 143.4, 143.3, 136.7, 132.7, 130.1, 129.9, 129.8, 128.5, 121.3, 119.2, 116.2, 110.0, 103.9, 69.3; LC-MS (ESI) 459.1 (M + H).

#### 3.1.5. (*E*)-*N*-(4-(3-(2-Hydroxy-6-((4-(trifluoromethoxy)benzyl)oxy)phenyl)-3-oxoprop-1-en-1-yl)phenyl)acetamide (21-5)

^1^H-NMR (400 MHz, CDCl_3_): δ 13.3 (s, 1H), 7.72 (s, 2H), 7.54–7.35 (m, 5H), 7.34–7.09 (m, 5H), 6.67 (d, *J* = 8.1 Hz, 1H), 6.52 (d, *J* = 8.0 Hz, 1H), 5.14 (s, 3H), 2.21 (s, 3H); ^13^C-NMR (100 MHz, DMSO-*d*_6_): d 193.2, 168.9, 165.3, 160.1, 146.5, 145.1, 137.7, 136.9, 130.8, 129.3, 129.0, 128.8, 121.5, 114.5, 114.0, 110.5, 105.3, 69.8, 24.0; LC-MS (ESI) 472.1 (M + H).

#### 3.1.6. (*E*)-1-(2-(Benzyloxy)-6-hydroxyphenyl)-3-(4-ethylphenyl)prop-2-en-1-one (21-6)

^1^H-NMR (400 MHz, CDCl_3_): δ 13.6 (s, 1H), 7.76 (dd, *J* = 15.6, 40.9 Hz, 2H), 7.50–7.27 (m, 6H), 7.01 (s, 4H), 6.61 (d, *J* = 8.3 Hz, 1H), 6.46 (d, *J* = 8.2 Hz, 1H), 5.04 (s, 2H), 2.61 (q, *J* = 7.6 Hz, 2H), 1.21 (t, *J* = 7.6 Hz, 3H); ^13^C-NMR (100 MHz, DMSO-*d*_6_): d 194.7, 159.2, 158.2, 147.3, 144.3, 137.0, 133.2, 129.0, 128.3, 127.8, 115.5, 110.0, 103.9, 70.5, 28.6, 15.7; LC-MS (ESI) 359.2 (M + H).

#### 3.1.7. (*E*)-1-(2-(Benzyloxy)-6-hydroxyphenyl)-3-(4-chlorophenyl)prop-2-en-1-one (21-7)

^1^H-NMR (400 MHz, CDCl_3_): δ 13.5 (s, 1H), 7.77 (d, *J* = 15.6 Hz, 1H), 7.61 (d, *J* = 15.6 Hz, 1H), 7.54–7.29 (m, 6H), 7.12 (d, *J* = 8.0 Hz, 2H), 6.91 (d, *J* = 8.0 Hz, 2H), 6.62 (d, *J* = 8.2 Hz, 1H), 6.49 (d, *J* = 8.1 Hz, 1H), 5.05 (s, 2H); ^13^C-NMR (100 MHz, DMSO-*d*_6_): d 194.5, 159.0, 158.2, 142.4, 137.0, 135.4, 133.8, 130.5, 129.4, 128.8, 128.4, 128.3, 109.9, 104.0, 70.5; LC-MS (ESI) 365.1 (M + H).

#### 3.1.8. (*E*)-1-(2-(Benzyloxy)-6-hydroxyphenyl)-3-(4-methoxyphenyl)prop-2-en-1-one (21-8)

^1^H-NMR (400 MHz, CDCl_3_): δ 14.00 (s, 1H), 7.77 (dd, *J* = 15.6, 27.5 Hz, 2H), 7.52–7.45 (m, 2H), 7.44–7.32 (m, 4H), 7.32–7.14 (m, 1H), 7.07 (d, *J* = 8.7 Hz, 2H), 6.73 (d, *J* = 8.7 Hz, 2H), 6.45 (d, *J* = 8.5 Hz, 1H), 5.08 (s, 2H), 3.97 (s, 3H); ^13^C-NMR (100 MHz, DMSO-*d*_6_): d 194.5, 161.7, 159.1, 158.1, 144.3, 137.1, 133.0, 130.8, 128.9, 128.4, 128.3, 127.4, 126.3, 115.6, 114.9, 109.9, 103.9, 70.4, 55.8; LC-MS (ESI) 360.8 (M^+^).

#### 3.1.9. (*E*)-4-(3-(2-(Benzyloxy)-6-hydroxyphenyl)-3-oxoprop-1-en-1-yl)benzoic acid (21-9)

^1^H-NMR (400 MHz, CDCl_3_): δ 7.96–7.88 (m, 2H), 7.74–7.67 (m, 1H), 7.54–7.35 (m, 6H), 7.33–7.08 (m, 5H), 6.67 (d, *J* = 8.0 Hz, 1H), 6.57 (d, *J* = 8.3 Hz, 1H), 5.14 (s, 2H); ^13^C-NMR (100 MHz, DMSO-*d*_6_): d 194.6, 158.9, 158.0, 143.1, 137.4, 137.1, 133.1, 130.2, 129.9, 128.8, 128.6, 128.5, 128.3, 128.2, 115.7, 110.0, 103.9, 70.4; LC-MS (ESI) 374.9 (M^+^).

#### 3.1.10. (*E*)-1-(2-(Benzyloxy)phenyl)-3-(4-ethylphenyl)prop-2-en-1-one (21-10)

^1^H-NMR (400 MHz, CDCl_3_): δ 12.9 (s, 1H), 10.0 (s, 1H), 7.96–7.88 (m, 2H), 7.81 (d, *J* = 8.1 Hz, 2H), 7.67–7.57 (m, 3H), 7.50 (t, *J* = 8.5 Hz, 1H), 7.36 (d, *J* = 8.0 Hz, 2H), 7.30–7.24 (m, 3H), 7.03 (d, *J* = 8.4 Hz, 1H), 7.00–6.87 (m, 1H), 2.78–2.66 (m, 3H), 1.27 (ddd, *J* = 2.2, 7.6, 7.6 Hz, 3H); ^13^C-NMR (100 MHz, CDCl_3_) d 192.4, 157.4, 146.9, 143.2, 136.3, 133.0, 132.7, 130.8, 129.6, 128.6, 128.5. 128.3, 127.5, 126.5, 121.2, 113.0, 70.7, 28.9, 15.4; LC-MS (ESI) 343.2 (M + H).

#### 3.1.11. (*E*)-1-(2-(Benzyloxy)phenyl)-3-(4-chlorophenyl)prop-2-en-1-one (21-11)

^1^H-NMR (400 MHz, CDCl_3_): δ 7.77 (dd, *J* = 1.6, 7.6 Hz, 1H), 7.59 (d, *J* = 16 Hz, 1H), 7.53–7.38 (m, 4H), 7.35–7.20 (m, 8H), 7.11–7.02 (m, 2H), 5.17 (s, 2H); ^13^C-NMR (100 MHz, CDCl_3_) d 192.8, 161.1, 145.2, 136.7, 135.5, 133.5, 133.2, 131.1, 129.9, 128.9, 128.7, 127.6, 127.1, 126.1, 122.3, 118.7, 117.2, 70.6; LC-MS (ESI) 349.1 (M + H).

#### 3.1.12. (*E*)-1-(2-(Benzyloxy)phenyl)-3-(4-methoxyphenyl)prop-2-en-1-one (21-12)

^1^H-NMR (400 MHz, CDCl_3_): δ 7.72 (dd, *J* = 2.0, 7.7 Hz, 1H), 7.61 (d, *J* = 15.8 Hz, 1H), 7.48–7.40 (m, 4H), 7.36–7.34 (m, 2H), 7.31–7.26 (m, 3H), 7.08–7.04 (m, 2H), 6.83 (d, *J* = 8.8 Hz, 2H), 5.17 (s, 2H), 3.84 (s, 3H); ^13^C-NMR (100 MHz, CDCl_3_): 192.3, 161.3, 157.4, 136.4, 132.9, 130.8, 130.1, 129.7, 128.6, 128.0, 127.9, 127.5, 125.2, 121.1, 114.2, 113.0, 70.7, 55.4; LC-MS (ESI) 345.2 (M + H).

#### 3.1.13. (*E*)-4-(3-(2-(Benzyloxy)phenyl)-3-oxoprop-1-en-1-yl)benzoic acid (21-13)

^1^H-NMR (400 MHz, CDCl_3_): δ 8.01 (d, *J* = 8.4 Hz, 2H), 7.81 (dd, *J* = 1.9, 7.7 Hz, 1H), 7.63 (d, *J* = 5.3 Hz, 2H), 7.52 (td, J = 2.0, 8.8 Hz, 1H), 7.45–7.43 (m, 2H), 7.40 (d, *J* = 8.4 Hz, 2H), 7.35–7.31 (m, 3H), 7.12–7.07 (m, 2H), 5.18 (s, 2H), 2.18 (s, 1H); ^13^C-NMR (100 MHz, CDCl_3_): d 192.8, 169.3, 161.1, 145.2, 140.4, 136.7, 135.5. 131.3, 130.2, 129.4, 129.0, 128.7, 127.6, 127.1. 126.1, 121.5, 118.7, 117.1, 70.2; LC-MS (ESI) 358.1 (M^+^).

#### 3.1.14. (*E*)-*N*-(4-(3-(2-(Benzyloxy)phenyl)-3-oxoprop-1-en-1-yl)phenyl)acetamide (21-14)

^1^H-NMR (400 MHz, CDCl_3_): δ 7.75-7.67 (m, 2H), 7.46–7.39 (m, 4H), 7.32–7.25 (m, 4H), 7.12–7.03 (m, 3H), 6.76 (d, *J* = 8.4 Hz, 1H), 6.59 (d, *J* = 8.5 Hz, 2H), 5.16 (s, 2H), 4.06-3.94 (m, 3H); ^13^C-NMR (100 MHz, CDCl_3_): d 192.8, 168.9, 162.1, 145.0, 137.7, 136.8, 135.4, 131.3, 130.8, 129.0, 128.7, 127.4, 127.0, 126.1, 121.5. 120.5, 114.7, 113.7, 70.4, 29.1; LC-MS (ESI) 372.2 (M + H).

#### 3.1.15. (*E*)-3-(4-Ethylphenyl)-1-(2-hydroxyphenyl)prop-2-en-1-one (21-15)

^1^H-NMR (400 MHz, CDCl_3_): δ 12.89 (s, 1H), 7.93–7.88 (m, 3H), 7.63–7.57 (m, 3H), 7.48 (t, *J* = 8.6 Hz, 1H), 7.25 (s, 1H), 7.02 (d, *J* = 8.3 Hz, 1H), 6.93 (t, *J* = 8.0 Hz, 1H), 2.69 (1, *J* = 5.7 Hz, 2H), 1.26 (t, *J* = 5.1 Hz, 3H); ^13^C-NMR (100 MHz, CDCl_3_): d 193.8, 163.6, 147.9, 145.6, 136.3, 132.2, 130.7, 129.6. 128.8, 128.6, 119.1, 118.9, 118.8, 118.6, 118.4, 28.9, 15.3; LC-MS (ESI) 253.1 (M + H).

#### 3.1.16. (*E*)-1-(2-Hydroxyphenyl)-3-(4-methoxyphenyl)prop-2-en-1-one (21-16)

^1^H-NMR (400 MHz, CDCl_3_): δ 12.98 (s, 1H), 7.89–7.87 (m, 2H), 7.57 (d, *J* = 8.8 Hz, 2H), 7.50–7.43 (m, 2H), 7.00 (dd, *J* = 0.8, 8.4 Hz, 1H), 6.92–6.88 (m, 3H), 3.81 (s, 3H); ^13^C-NMR (100 MHz, CDCl_3_): d 193.7, 163.6, 163.5, 162.0, 145.3, 136.1, 130.5, 129.5, 127.4, 127.3, 120.1, 118.7, 118.6, 117.6, 114.5, 55.4; LC-MS (ESI) 255.1 (M + H).

#### 3.1.17. (*E*)-*N*-(4-(3-(2-Hydroxyphenyl)-3-oxoprop-1-en-1-yl)phenyl)acetamide (21-17)

^1^H-NMR (400 MHz, CDCl_3_): δ 12.88 (s, 1H), 9.92 (s, 1H), 7.94-7.90 (m, 1H), 7.85 (d, J = 8.9 Hz, 2H), 7.72-7.42 (m, 5H), 7.04 (t, *J* = 7.9 Hz, 1H), 6.95 (t, *J* = 7.6 Hz, 1H), 2.23 (s, 3H); ^13^C-NMR (100 MHz, CDCl_3_): d 193.9, 162.9, 146.9, 135.49, 135.47, 131.0, 129.6, 118.7, 117.7, 114.3, 114.2, 114.15, 114.13, 114.1, 29.1; LC-MS (ESI) 282.1 (M + H).

#### 3.1.18. (*E*)-1-(2-(Cyclohexylmethoxy)phenyl)-3-(4-ethylphenyl)prop-2-en-1-one (21-18)

^1^H-NMR (400 MHz, CDCl_3_): δ 7.68–7.62 (m, 2H), 7.50 (d, *J* = 8.1 Hz, 2H), 7.48 (s, 1H), 7.45–7.40 (m, 1H), 7.45–7.40 (m, 1H), 7.21 (d, *J* = 8.1 Hz, 2H), 7.00 (td, *J* = 0.6, 7.5 Hz, 1H), 6.94 (d, *J* = 8.3 Hz, 1H), 3.83 (d, *J* = 5.8 Hz, 2H), 2.66 (q, *J* = 7.6 Hz, 1H), 1.82–1.57 (m, 6H), 1.24 (t, *J* = 7.6Hz, 3H), 1.19–1.01 (m, 5H); ^13^C-NMR (100 MHz, CDCl_3_): d 192.8, 158.0, 145.1, 143.4, 135.1, 132.4, 192.8, 128.5, 125.7, 120.8, 118.7, 114.9, 74.8, 38.4, 30.0, 28.2, 26.2, 25.8, 14.5; LC-MS (ESI) 349.2 (M + H).

#### 3.1.19. (*E*)-3-(4-Chlorophenyl)-1-(2-(cyclohexylmethoxy)phenyl)prop-2-en-1-one (21-19)

^1^H-NMR (400 MHz, CDCl_3_): δ 7.78–7.74 (m, 4H), 7.69 (d, *J* = 8.4Hz, 2H), 7.65–7.61 (m, 2H), 7.20 (d, *J* = 8.4 Hz, 2H), 3.84 (s, 2H), 1.84–1.58 (m, 6H), 1.20–1.02 (m, 5H); ^13^C-NMR (100 MHz, CDCl_3_): d 192.8, 158.0, 143.1, 135.0, 133.5, 133.6, 131.0, 130.0, 128.9, 125.7, 120.8, 118.7, 115.0, 74.8, 38.4, 30.0, 28.2, 26.2, 25.8; LC-MS (ESI) 355.1 (M + H).

#### 3.1.20. (*E*)-4-(3-(2-(Cyclohexylmethoxy)phenyl)-3-oxoprop-1-en-1-yl)benzoic acid (21-20)

^1^H-NMR (400 MHz, CDCl_3_): δ 8.13 (d, *J* = 8.3 Hz, 2H), 7.73–7.46 (m, 6H), 7.03 (t, *J* = 7.5 Hz, 1H), 6.98 (d, *J* = 8.5 Hz, 1H), 3.87 (d, *J* = 5.8 Hz, 2H), 1.83–1.58 (m, 6H), 1.17–1.00 (m, 5H); ^13^C-NMR (100 MHz, CDCl_3_): d 193.0, 168.2, 157.9, 145.0, 140.4, 135.1, 131.2, 130.2, 129.3, 129.0, 125.7, 120.8, 118.7, 114.5, 75.0, 38.5, 30.0, 26.0, 25.8; LC-MS (ESI) 365.1 (M + H).

#### 3.1.21. (*E*)-*N*-(4-(3-(2-(Cyclohexylmethoxy)phenyl)-3-oxoprop-1-en-1-yl)phenyl)acetamide (21-21)

^1^H-NMR (400 MHz, CDCl_3_): δ 8.42 (s, 1H), 7.65–7.57 (m, 4H), 7.51 (d, *J* = 8.6 Hz, 2H), 7.46–7.39 (m, 2H), 7.00 (t, *J* = 7.4 Hz, 1H), 6.95 (d, *J* = 8.4 Hz, 1H), 3.82 (d, *J* = 5.8 Hz, 2H), 2.17 (s, 3H), 1.81–1.56 (m, 6H), 1.16–1.01 (m, 5H); ^13^C-NMR (100 MHz, CDCl_3_): d 192.9, 167.0, 156.8, 143.2, 137.7, 135.1, 132.2, 130.8, 129.5, 125.4, 121.6. 120.8, 118.5, 114.1, 74.9, 38.4, 30.0, 26.2, 25.9. 24.5; LC-MS (ESI) 378.2 (M + H).

### 3.2. Biological Assays

#### 3.2.1. Reagents

α-MSH, kojic acid, arbutin, l-DOPA, and synthetic melanin were obtained from Sigma-Aldrich (St. Louis, MO, USA). Primary antibodies specific for tyrosinase, TRP1, TRP2, and MITF were obtained from Novus Biologicals, and phospho-ERK1/2, ERK1/2, phospho-CREB, CREB, and GAPDH were obtained from Cell Signaling Technology (Danvers, MA, USA). All other chemicals were purchased from Sigma-Aldrich, unless otherwise noted.

#### 3.2.2. Cell Line and Culture Conditions

The mouse melanoma cell line B16F10 was obtained from the American Type Culture Collection (ATCC, Manassas, VA, USA) and was maintained in Dulbecco’s modified Eagle’s medium (DMEM, Hyclone, Pittsburgh, PA, USA) supplemented with 10% fetal bovine serum (FBS) and 1% penicillin-streptomycin solution (Gibco BRL, Grand Island, NY, USA) at 37 °C in a humidified incubator suppling 5% CO_2_.

#### 3.2.3. Cellular Melanin Contents

Cells were seeded at a density of 2,500/well in 96-well culture plates and incubated for 24 h. The cells were incubated for 72 h with vehicle (0.1% dimethyl sulfoxide, DMSO) alone or with various concentrations of chalcone 21-21 in the absence or presence of α-MSH (10 nM). Melanin contents were determined in the cell-free culture medium by measuring the absorbance at 405 nm using a microplate reader. Results were calculated from a standard curve generated using synthetic melanin. Kojic acid and arbutin were used as positive controls.

#### 3.2.4. Cell Viability Assay

Cells were seeded at a density of 2500/well in 96-well culture plates. After overnight incubation, cells were treated with vehicle (0.1% DMSO) alone or with various concentrations of chalcone 21-21 for appropriate time intervals in the absence or presence of α-MSH (10 nM). Cell viability was measured by the absorbance at 450 nm in a microplate reader (Molecular Devices) after further incubation for 2–4 h at 37 °C, followed by the addition with EZ-CyTox Enhanced Cell Viability Assay Reagent (Daeil Lab Service) according to the manufacturer’s protocols.

#### 3.2.5. Zymography

Cells were incubated for 72 h with vehicle (0.1% DMSO) alone or various concentrations of chalcone 21-21 in the absence or presence of α-MSH (10 nM). Whole-cell lysates were prepared on ice for 1 h using a sodium phosphate lysis buffer containing 0.1 M sodium phosphate buffer (pH 6.8), 1% Triton X-100, 1 mM phenylmethylsulfonyl fluoride (PMSF) and phosphatase inhibitor cocktail (Thermo Scientific, Pittsburgh, PA, USA). Soluble protein samples were prepared by centrifugation at 13,000 rpm for 10 min at 4 °C and the samples were resolved with Laemmli sample buffer without β-mercaptoethanol by native SDS-PAGE. The gels were equilibrated and soaked in 100 mM sodium phosphate buffer (pH 6.8) containing 5 mM l-DOPA until colorimetric detection of tyrosinase.

#### 3.2.6. Western Blot Analysis

Whole-cell lysates were prepared on ice for 30 min using a lysis buffer containing 50 mM Tris-HCl (pH 7.4), 350 mM NaCl, 1% Triton X-100, 0.5% Nonidet P-40, 10% glycerol, 0.1% SDS, 1 mM EDTA, 1 mM EGTA, 1 mM Na_3_VO_4_, 1 mM PMSF, and phosphatase inhibitor cocktail (Thermo Scientific, Pittsburgh, PA, USA). The lysates were resolved by SDS-PAGE and transferred onto nitrocellulose membrane (Sigma-Aldrich, St. Louis, MO, USA). The membrane was blocked in a blocking buffer containing 5% skim-milk and subsequently incubated with appropriate primary antibodies for overnight at 4 °C. Blots were washed and incubated with horseradish peroxidase-conjugated secondary antibodies at room temperature for 2 h, and the signals were detected using an ECL reagent (SurModics, Eden Prairie, MN, USA).

#### 3.2.7. Semi-Quantitative RT-PCR

Total RNA was isolated using a TRIzol Reagent (Invitrogen, Carlsbad, CA, USA) and semi-quantitative reverse-transcription PCR (RT-PCR) was performed using an AccuPower RT-PCR PreMix (Bioneer, Daejeon, Korea) according to the manufacturer’s protocol. Briefly, total RNA was reverse-transcribed on a thermal cycler programmed at 42 °C for 1 h and PCR was performed 30 cycles at 94 °C for 30 s, 58 °C for 30 s and 72 °C for 30 s, and ending with one cycle at 72 °C for 5 min. The amplicons were resolved on agarose gels with DNA SafeStain (Lamda Biotech, Ballwin, MO, USA) and visualized using a Gel Doc 2000 imaging system (Bio-Rad Laboratories, Hercules, CA, USA). The primer sequences were described in Table 2.

#### 3.2.8. Transfection and Promoter Assay

Cells were transiently transfected with the pTyr-luciferase, pMITF-luciferase, or pCRE-luciferase construct together with the pRL-TK vector (Promega, Madison, WI, USA) using a Lipofectamine^®^ PLUS^TM^ reagent (Life Technologies, Carlsbad, CA, USA) as described previously [13]. After 24 h, cells were incubated for 24 h with vehicle (0.1% DMSO) alone or with various concentrations of chalcone 21-21 in the absence or presence of α-MSH (10 nM). Luciferase activities were measured using a Dual-Glo luciferase assay system (Promega) and firefly luciferase activity was normalized to that of *Renilla* luciferase.

#### 3.2.9. Statistics

Data are presented as mean ± standard deviation (SD) and statistical differences were determined using one-way analysis of variance, followed by Dunnett’s multiple comparison tests. Statistical significance was considered to be significant at *p* < 0.05.

## 4. Conclusions

Melanin is synthesized in the melanosomes of melanocytes, and is transferred to and eventually distributed in the epidermis [24]. Although it has a skin protective role, abnormal accumulation can lead to various skin disorders, indicating a need for the development of anti-melanogenic agents. In this study, we synthesized a series of derivatives of chalcone 21 to optimize and increase its previously reported anti-melanogenic activity [13]. Activity was evaluated in α-MSH-stimulated B16F10 cells by measuring cellular melanin production. Based on the SAR, moderate to strong biological activity with weak cytotoxicity was observed when an acetamido group, carboxyl group, or chloride atom was located at the R_3_ position. Among the derivatives, chalcone 21-21 had the strongest inhibitory effect on cellular melanin production, with an IC_50_ value of 0.54 μM. This is more potent than chalcone 21 and the known anti-melanogenic agents kojic acid and arbutin, which had IC_50_ values of 4.9, 38.5, and 148.4 μM, respectively.

Melanin biosynthesis is tightly regulated by several complex signaling networks. Among the signaling networks, tyrosinase, TPR1, TPR2, MITF, CREB, and ERK1/2 are well-known key regulators of melanin synthesis [1,6,19,20,21,22,23]. Chalcone 21-21 inhibited the expression and activity of tyrosinase, which catalyzes the rate-limiting step in melanin biosynthesis by catalyzing the conversion of L-tyrosine to L-DOPA by hydroxylation. In addition, chalcone 21-21 inhibited the expression and transcriptional activity of MITF, a master transcription factor that drives the expression of target genes encoding tyrosinase, TRP1, and TRP2 [23], this inhibition resulted in reducing the expression of MITF-regulated target genes. Furthermore, chalcone 21-21 inhibited the phosphorylation of ERK1/2 and CREB, as well as the transcriptional activity of CREB. ERK/CREB signaling plays an important role in regulating the transcription and phosphorylation of MITF. Our results indicate that the anti-melanogenic activity of chalcone 21-21 is associated with the inhibition of ERK/CREB signaling, thereby targeting the expression of MITF and MITF-regulated genes including those encoding tyrosinase, TRP1, and TRP2 (Figure 6). Finally, our results demonstrate that chalcone-21-21 is an effective anti-melanogenic agent and may be beneficial in the treatment of aesthetic problems and hyperpigmentation disorders.

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
