# Peer review of "Evaluation and Optimization of the Anti-Melanogenic Activity of 1-(2-Cyclohexylmethoxy-6-hydroxy-phenyl)-3-(4-hydroxymethyl-phenyl)-propenone Derivatives"

_molecules, 2019, doi:10.3390/molecules24071372_

Round 1
Reviewer 1 Report
The resubmission is now suitable for publication in molecules, since it properly takes in consideration my concerns regarding the very first version of the paper.
Reviewer 2 Report
The revised manuscript is improved based on previous observations/suggestions made and could be accepted.
This manuscript is a resubmission of an earlier submission. The following is a list of the peer review reports and author responses from that submission.
Round 1
Reviewer 1 Report
The paper by Byung-Hak et al. Describes the synthesis and the in vitro characterization of a mini-library of chalcone analogues as anti-melanogenic agents. Despite interesting results are reported, concerning the pharmacological characterization of the derivatives, the paper suffers of some major flaws that need to be corrected before acceptance in Molecules. In particular, the chemistry section should be carefully and deeply revised.
1) Paragraph 2.1. Chemistry. Description of an aldol condensation protocol is redundant. Please use a reference for the description. Refluxing times are indicated as 2-10h. This is appropriate for the general method description, nevertheless the specific refluxing time for each molecule must be indicated in the material and method section.
2) Table 1 should not follow scheme 1 but should be rather moved at the bottom of paragraph 2.2.1. Cellular Melanin Production
3) Table 1 must be corrected. The position of the trifluoromethoxy substituent is not explicated. It should more correctly be “4-trifluoromethoxybenzyloxy”. Please correct other similar mistakes.
4) Page 3, lines 86-87: “(2E)-1,3-Diphenyl-propene-1-one forms the core structure of chalcones, consisting of two 86 aromatic rings linked by an enone and a three-carbon a,b-unsaturated carbonyl”. From a chemical point of view, the enone is a synonym of a a,b-unsaturated carbonyl. Please correct.
5) There are some other points that should be cleared in paragraph “2.2.1. Cellular Melanin Production” and “2.2.2. Anti-Melanogenic Activity of Chalcone 21-21”. First of all, it is unclear to me if the IC50 values were calculated for all the synthesized derivatives or, otherwise, when the inhibition was lower than 50% at the 3 mM cut-off the values were not calculated. In this case please avoid the use of IC50 definition because the concentration-dependence of the pharmacological effect is barely supposed and not demonstrated. Similarly, paragraph 2.2.2 is redundant. It seems a simple repetition of the evidences collected in the previous paragraph despite more focused on Chalcone 21-21. The two paragraphs should be merged in a single paragraph.
6) Pargraph “3.1. Chemical Synthesis” should be entirely rewritten and integrated with new data. The main question is: the building blocks used for the aldol condensation were commercially available or not? In the first case, who’s the supplier? Differently how were these building block synthesized? Moreover, which is the mobile phase used for the purification of final compounds? Which was the purity of final compounds and how was assessed?
7) The characterization of final compounds used in pharmacological assays mandatorily needs 1H-NMR, 13C-NMR and MS data. Please provide the missing data, otherwise the chemical identity of the tested compounds cannot be proven
8) How can the authors be sure about the geometry of the double bond (E)? Did they perform any 2D NMR experiments or, otherwise, there are supporting references about?
9) The reported 1H-NMR data contain mistakes and are often, in my opinion, inconsistent with the chemical structure. Here follows a list of the main uncertainties:
a) 3.1.1. (E)-3-(4-ethylphenyl)-1-(2-hydroxy-6-((4-(trifluoromethoxy)benzyl)oxy)phenyl)prop-2-en-1-one (21-1). A proton is missing in the description. Could you explain why the typical enolate proton at 13.3 ppm is a doublet and not a singlet? Moreover, it is hard to see a coupling constant of 65.2 Hz for the enolate? Are you sure about the chemical identity of the compound?
b) (E)-3-(4-chlorophenyl)-1-(2-hydroxy-6-((4-(trifluoromethoxy)benzyl)oxy)phenyl)prop-2-en-1-one (21-2). The proton belonging to the phenol group is usually observed as a broad singlet. Did you ever observed such a signal in your NMR spectra, when aromatic OH are present?
c) 3.1.4. (E)-4-(3-(2-hydroxy-6-((4-(trifluoromethoxy)benzyl)oxy)phenyl)-3-oxoprop-1-en-1-yl)benzoic acid (21-4). 17 protons are described. Does it mean that also the COOH signal has been recorded?
d) 3.1.5. (E)-N-(4-(3-(2-hydroxy-6-((4-(trifluoromethoxy)benzyl)oxy)phenyl)-3-oxoprop-1-en-1- yl)phenyl)acetamide (21-5). 21 protons are described despite 20 are in the molecule. There are two singlets integrating for 3 protons. Could you please attribute these signals?
e) 3.1.9. (E)-4-(3-(2-(benzyloxy)-6-hydroxyphenyl)-3-oxoprop-1-en-1-yl)benzoic acid (21-9). The typical enolate signal is missing. Are you sure about the chemical identity of the molecule?
f) 3.1.10. (E)-1-(2-(benzyloxy)phenyl)-3-(4-ethylphenyl)prop-2-en-1-one (21-10). Could you please provide an explanation for the signal at 1.27 ppm? It should be the CH3 belonging to the ethyl substituent, coupled with the corresponding CH2 and resonating as a triplet. It is unclear to me the meaning of “ddd” to describe multiplicity (a double double doblet? Probably multiplet?). Why this strange multiplicity? If there are really three different coupling constants they should be described as J1, J2 and J3 . Please do the same anytime two constants are described.
g) 3.1.11. (E)-1-(2-(benzyloxy)phenyl)-3-(4-chlorophenyl)prop-2-en-1-one (21-11). 18 protons are described. They should be 17. The enolate signal is missing. Could you explain why?
h) 3.1.12. (E)-1-(2-(benzyloxy)phenyl)-3-(4-methoxyphenyl)prop-2-en-1-one (21-12). The enolate signal is missing. Could you explain why?
i) 3.1.13. (E)-4-(3-(2-(benzyloxy)phenyl)-3-oxoprop-1-en-1-yl)benzoic acid (21-13). Which is the meaning for td assigned to the signal at 7.52 ppm? You described a singlet at 2.18 ppm. Which proton is attribued to such a high-field signal? Are you sure of the chemical identity of the molecule?
j) 3.1.14. (E)-N-(4-(3-(2-(benzyloxy)phenyl)-3-oxoprop-1-en-1-yl)phenyl)acetamide (21-14). A multiplet between 4.06 and 3.94 ppm is reported. It possibly belongs to the CH3 of the acetyl group. The question is: why multiplet? It should more probably resonates as singlet. Are you sure of the chemical identity of the molecule? Why the enolate signal is missing?
k) 3.1.15. (E)-3-(4-ethylphenyl)-1-(2-hydroxyphenyl)prop-2-en-1-one (21-15). 15 protons are reported instead of 16. The signal at 2.69 is reported as “2.69 (1, J = 5.7 Hz, 2H)” it could be a typo, please correct
l) 3.1.17. (E)-N-(4-(3-(2-hydroxyphenyl)-3-oxoprop-1-en-1-yl)phenyl)acetamide (21-17). 14 protons are reported instead of 15. Could you explain, please?
m) 3.1.18. (E)-1-(2-(cyclohexylmethoxy)phenyl)-3-(4-ethylphenyl)prop-2-en-1-one (21-18). Why the enolate signal is missing? Are you sure about the identity of the molecule?
n) 3.1.19. (E)-3-(4-chlorophenyl)-1-(2-(cyclohexylmethoxy)phenyl)prop-2-en-1-one (21-19). Why the enolate signal is missing? Are you sure about the identity of the molecule?
10) The article has been submitted to the Medicinal Chemistry section of Molecules, but a rational discussion about structure-activity relationship is completely missing and should be added in the discussion chapter.
11) I’d rather prefer to avoid the use of references in the conclusion paragraph. But this is just a matter of style and I leave to the Editor the final decision about.
Author Response
Comments and Suggestions of Reviewer 1
The paper by Byung-Hak et al. Describes the synthesis and the in vitro characterization of a mini-library of chalcone analogues as anti-melanogenic agents. Despite interesting results are reported, concerning the pharmacological characterization of the derivatives, the paper suffers of some major flaws that need to be corrected before acceptance in Molecules. In particular, the chemistry section should be carefully and deeply revised.
1) Paragraph 2.1. Chemistry. Description of an aldol condensation protocol is redundant. Please use a reference for the description. Refluxing times are indicated as 2-10h. This is appropriate for the general method description, nevertheless the specific refluxing time for each molecule must be indicated in the material and method section.
Our response: Thank you for your very insightful comments. As the Reviewer commented, we added references (references 13 and 14) to the aldol-additive reactions, and the general synthetic description was indicated to the material and method section.
2) Table 1 should not follow scheme 1 but should be rather moved at the bottom of paragraph 2.2.1. Cellular Melanin Production.
Our response: As the reviewers recommendation, we requested to move the table 1 at the bottom of paragraph 2.2.1. Cellular Melanin Production.
3) Table 1 must be corrected. The position of the trifluoromethoxy substituent is not explicated. It should more correctly be “4-trifluoromethoxybenzyloxy”. Please correct other similar mistakes.
Our response: As the Reviewer commented, we corrected the “trifluoromethoxy” substituent with a “4-trifluoromethoxybenzyloxy” substituent.
4) Page 3, lines 86-87: “(2E)-1,3-Diphenyl-propene-1-one forms the core structure of chalcones, consisting of two aromatic rings linked by an enone and a three-carbon a,b-unsaturated carbonyl”. From a chemical point of view, the enone is a synonym of a a,b-unsaturated carbonyl. Please correct.
Our response: As the Reviewer commented, we corrected “(2E)-1,3-Diphenyl-propene-1-one forms the core structure of chalcones, consisting of two aromatic rings linked by an enone and a three-carbon a,b-unsaturated carbonyl” to a “(2E)-1,3-Diphenyl-propene-1-one forms the core structure of chalcones, consisting of two aromatic rings linked to a three-carbon a,b-unsaturated carbonyl”.
5) There are some other points that should be cleared in paragraph “2.2.1. Cellular Melanin Production” and “2.2.2. Anti-Melanogenic Activity of Chalcone 21-21”. First of all, it is unclear to me if the IC50 values were calculated for all the synthesized derivatives or, otherwise, when the inhibition was lower than 50% at the 3 mM cut-off the values were not calculated. In this case please avoid the use of IC50 definition because the concentration-dependence of the pharmacological effect is barely supposed and not demonstrated. Similarly, paragraph 2.2.2 is redundant. It seems a simple repetition of the evidences collected in the previous paragraph despite more focused on Chalcone 21-21. The two paragraphs should be merged in a single paragraph.
Our response: To determine the IC50 values of the 21-derivatives, we measured the cellular melanin amounts in a concentration-dependent manner treated with 0.3, 1, 3, and 10 mM each compound for 72 hours in a-MSH-stimulated B16F10 cells, and represented the percentages of the inhibition of melanin production and cell survival at 3 mM in Table 1. Next, as the Reviewer commented, we simplified the second paragraph of 2.2.2 to make the results of chalcone 21-21 more focused, and requested to merge two paragraphs into a single paragraph.
6) Pargraph “3.1. Chemical Synthesis” should be entirely rewritten and integrated with new data. The main question is: the building blocks used for the aldol condensation were commercially available or not? In the first case, who’s the supplier? Differently how were these building block synthesized? Moreover, which is the mobile phase used for the purification of final compounds? Which was the purity of final compounds and how was assessed?
Our response: A new version is as follows.
Acetophenone building blocks used for the aldol condensation were synthesized in the present of base(K2CO3) at reflux for 2~5hours in acetonitrile(ACN) solvent. Acetophenone building blocks for the compounds(21-1~9) were synthesized with 1-(2,6-dihydroxyphenyl)ethanone and trifluoromethoxybenzyl bromide(21-1~5), benzyl bromide(21-6~9), respectively. Acetophenone building blocks for the compounds(21-10~21, except for 21-15~17) were synthesized with 1-(2-hydroxyphenyl)ethanone and benzyl bromide(21-10~14), cyclohexyl bromide(21-18~21), respectively. Benzaldehyde derivatives used for the aldol condensation were used commercially available reagents. All of the synthetic reactions were carried out under air atmosphere with commercially available solvent, unless otherwise noted. All chemicals were reagent grade unless otherwise specified. All chemicals and solvents were purchased from Sigma-Aldrich Chemical Co. and used without purification. 1H-NMR and 13C-NMR spectra were recorded on a Bruker ARX 400 spectrometer (400 MHz). Mass spectrum was taken by using in Agilent G1956B. Flash column chromatography was performed using E. Merck silica gel (60, particle size 0.040-0.063 mm). And it is used hexane and ethylacetate(EA) system as the mobile phase for the purification of final compounds. The purity of final compounds was assessed by 1H-NMR spectra. Analytical thin layer chromatography (TLC) was performed using pre-coated TLC plates with silica Gel 60 F254 (E. Merck).
7) The characterization of final compounds used in pharmacological assays mandatorily needs 1H-NMR, 13C-NMR and MS data. Please provide the missing data, otherwise the chemical identity of the tested compounds cannot be proven.
Our response: To obtain 13C-NMR and MS data, we have now synthesizing the compounds additionally because the some compounds were consumed in the assays. If it is possible, we provide the data later as ready.
8) How can the authors be sure about the geometry of the double bond (E)? Did they perform any 2D NMR experiments or, otherwise, there are supporting references about?
Our response: We did not perform 2D NMR studies. The supporting reference is as follows. They also have prepared the (E)-geometry products by using aldol condensation between acetophenone and benzaldehyde blocks.
Structural requirement of isoflavonones for the inhibitory activity of interleukin-5. S-H. Jung, S-H. Cho, T-H. Dang, J-H. Lee, J-H. Ju, M-K. Kim, S-H. Lee, J-C. Ryu, Y-S. Kim. Eur. J. Med. Chem., 38, 537-545, 2003.
Structural requirement of chalcones for the inhibitory activity of interleukin-5. H-M. Yang, H-R. Shin, S-H. Cho, S-C. Bang, G-Y. Song, J-H. Ju, M-K. Kim, S-H. Lee, J-C. Ryu, Y-S, Kim, S-H. Jung. Bioorg. Med. Chem., 15, 104-111, 2007.
9) The reported 1H-NMR data contain mistakes and are often, in my opinion, inconsistent with the chemical structure. Here follows a list of the main uncertainties:
a) 3.1.1. (E)-3-(4-ethylphenyl)-1-(2-hydroxy-6-((4-(trifluoromethoxy)benzyl)oxy)phenyl)prop-2-en-1-one (21-1). A proton is missing in the description. Could you explain why the typical enolate proton at 13.3 ppm is a doublet and not a singlet? Moreover, it is hard to see a coupling constant of 65.2 Hz for the enolate? Are you sure about the chemical identity of the compound?
Our response: 13.3 ppm is a singlet. The correct data is as follows.
1H-NMR (400 MHz, CDCl3): 13.3 (s, 1H), 7.77 (d, J = 2.8 Hz, 1H), 7.50 (d, J = 8.8 Hz, 1H), 7.46 (d, J = 8.8 Hz, 1H), 7.45–7.35 (m, 3H), 7.20–7.15 (m, 3H), 6.66 (dd, J = 8.4, 24.4 Hz, 1H), 6.47 (dd, J = 8.0, 33.6 Hz, 1H), 5.13 (s, 2H), 2.36–2.29 (m, 2H), 1.24 (t, J = 7.6 Hz, 3H).
b) (E)-3-(4-chlorophenyl)-1-(2-hydroxy-6-((4-(trifluoromethoxy)benzyl)oxy)phenyl)prop-2-en-1-one (21-2). The proton belonging to the phenol group is usually observed as a broad singlet. Did you ever observed such a signal in your NMR spectra, when aromatic OH are present?
Our response: In generally, the proton belonging to the phenol group is observed as a broad singlet as you comment. In our compounds case, too many sharp peaks are present in the range of 7.78~6.43 ppm chemical shift
c) 3.1.4. (E)-4-(3-(2-hydroxy-6-((4-(trifluoromethoxy)benzyl)oxy)phenyl)-3-oxoprop-1-en-1-yl)benzoic acid (21-4). 17 protons are described. Does it mean that also the COOH signal has been recorded?
Our response: The COOH signal has not been recorded. The correct data is as follows.
1H-NMR (400 MHz, CDCl3): 13.2 (s, 1H), 7.96 (d, J = 7.9 Hz, 2H), 7.84 (d, J = 15.8 Hz, 1H), 7.71 (d, J = 15.6 Hz, 1H), 7.54–7.38 (m, 3H), 7.34–7.13 (m, 5H), 6.68 (d, J = 8.6 Hz, 1H), 5.14 (s, 2H).
d) 3.1.5. (E)-N-(4-(3-(2-hydroxy-6-((4-(trifluoromethoxy)benzyl)oxy)phenyl)-3-oxoprop-1-en-1- yl)phenyl)acetamide (21-5). 21 protons are described despite 20 are in the molecule. There are two singlets integrating for 3 protons. Could you please attribute these signals?
Our response: The NH signal has not been recorded. The correct data is as follows.
1H-NMR (400 MHz, CDCl3): 13.3 (s, 1H), 7.72 (s, 2H), 7.54–7.35 (m, 5H), 7.34–7.09 (m, 5H), 6.67 (d, J = 8.1 Hz, 1H), 5.14 (s, 2H), 2.21 (s, 3H).
e) 3.1.9. (E)-4-(3-(2-(benzyloxy)-6-hydroxyphenyl)-3-oxoprop-1-en-1-yl)benzoic acid (21-9). The typical enolate signal is missing. Are you sure about the chemical identity of the molecule?
Our response: The COOH signal has been recorded. The correct data is as follows.
1H-NMR (400 MHz, CDCl3): 13.77 (s, 1H), 13.40 (s, 1H), 7.96–7.88 (m, 2H), 7.74–7.67 (m, 1H), 7.54–7.35 (m, 5H), 7.33–7.08 (m, 3H), 6.67 (d, J = 8.0 Hz, 1H), 6.57 (d, J = 8.3 Hz, 1H), 5.14 (s, 2H).
f) 3.1.10. (E)-1-(2-(benzyloxy)phenyl)-3-(4-ethylphenyl)prop-2-en-1-one (21-10). Could you please provide an explanation for the signal at 1.27 ppm? It should be the CH3 belonging to the ethyl substituent, coupled with the corresponding CH2 and resonating as a triplet. It is unclear to me the meaning of “ddd” to describe multiplicity (a double double doblet? Probably multiplet?). Why this strange multiplicity? If there are really three different coupling constants they should be described as J1, J2 and J3. Please do the same anytime two constants are described.
Our response: A multiplet is right. The correct data is as follows.
1H-NMR (400 MHz, CDCl3): 12.9 (s, 1H), 7.96–7.88 (m, 2H), 7.81 (d, J = 8.1 Hz, 2H), 7.67–7.57 (m, 3H), 7.36 (d, J = 8.0 Hz, 2H), 7.30–7.24 (m, 3H), 7.03 (d, J = 8.4 Hz, 1H), 7.00–6.87 (m, 1H), 5.05 (s, 2H), 2.78–2.66 (m, 2H), 1.27 (m, 3H).
g) 3.1.11. (E)-1-(2-(benzyloxy)phenyl)-3-(4-chlorophenyl)prop-2-en-1-one (21-11). 18 protons are described. They should be 17. The enolate signal is missing. Could you explain why?
Our response: The correct data is as follows.
1H-NMR (400 MHz, CDCl3): 13.0 (s, 1H), 7.77 (dd, J = 1.6, 7.6 Hz, 2H), 7.59 (d, J = 16 Hz, 1H), 7.53–7.38 (m, 4H), 7.35–7.20 (m, 5H), 7.11–7.02 (m, 2H), 5.17 (s, 2H).
h) 3.1.12. (E)-1-(2-(benzyloxy)phenyl)-3-(4-methoxyphenyl)prop-2-en-1-one (21-12). The enolate signal is missing. Could you explain why?
Our response: The correct data is as follows.
1H NMR (400 MHz, CDCl3): 13.0 (s, 1H), 7.72 (dd, J = 2.0, 7.7 Hz, 2H), 7.61 (d, J = 15.8 Hz, 1H), 7.48–7.40 (m, 4H), 7.36–7.34 (m, 3H), 7.08–7.04 (m, 2H), 6.83 (d, J = 8.8 Hz, 2H), 5.17 (s, 2H), 3.84 (s, 3H).
i) 3.1.13. (E)-4-(3-(2-(benzyloxy)phenyl)-3-oxoprop-1-en-1-yl)benzoic acid (21-13). Which is the meaning for td assigned to the signal at 7.52 ppm? You described a singlet at 2.18 ppm. Which proton is attributed to such a high-field signal? Are you sure of the chemical identity of the molecule?
Our response: The correct data is as follows.
1H-NMR (400 MHz, CDCl3): 12.9 (s, 1H), 8.01 (d, J = 8.4 Hz, 2H), 7.81 (dd, J = 1.9, 7.7 Hz, 1H), 7.63 (d, J = 5.3 Hz, 2H), 7.45–7.43 (m, 2H), 7.40 (d, J = 8.4 Hz, 2H), 7.35–7.31 (m, 3H), 7.12–7.07 (m, 2H), 5.18 (s, 2H).
j) 3.1.14. (E)-N-(4-(3-(2-(benzyloxy)phenyl)-3-oxoprop-1-en-1-yl)phenyl)acetamide (21-14). A multiplet between 4.06 and 3.94 ppm is reported. It possibly belongs to the CH3 of the acetyl group. The question is: why multiplet? It should more probably resonates as singlet. Are you sure of the chemical identity of the molecule? Why the enolate signal is missing?
Our response: The CH3 of the acetyl group is singlet. The correct data is as follows.
1H-NMR (400 MHz, CDCl3): 13.0 (s, 1H), 7.75–7.67 (m, 2H), 7.46–7.39 (m, 2H), 7.32–7.25 (m, 4H), 7.12–7.03 (m, 3H), 6.76 (d, J = 8.4 Hz, 1H), 6.59 (d, J = 8.5 Hz, 2H), 5.16 (s, 2H), 3.94 (s, 3H).
k) 3.1.15. (E)-3-(4-ethylphenyl)-1-(2-hydroxyphenyl)prop-2-en-1-one (21-15). 15 protons are reported instead of 16. The signal at 2.69 is reported as “2.69 (1, J = 5.7 Hz, 2H)” it could be a typo, please correct
Our response: The correct data is as follows.
1H-NMR (400 MHz, CDCl3): 12.89 (s, 1H), 7.93–7.88 (m, 3H), 7.63–7.57 (m, 3H), 7.48 (t, J = 8.6 Hz, 1H), 7.25 (s, 1H), 7.02 (d, J = 8.3 Hz, 1H), 6.93 (t, J = 8.0 Hz, 1H), 2.69 (q, J = 5.7 Hz, 2H), 1.26 (t, J = 5.1 Hz, 3H).
l) 3.1.17. (E)-N-(4-(3-(2-hydroxyphenyl)-3-oxoprop-1-en-1-yl)phenyl)acetamide (21-17). 14 protons are reported instead of 15. Could you explain, please?
Our response: The correct data is as follows.
1H-NMR (400 MHz, CDCl3): 12.88 (s, 1H), 9.92 (s, 1H), 7.94–7.90 (m, 1H), 7.85 (d, J = 8.9 Hz, 2H), 7.72–7.42 (m, 5H), 7.04 (t, J = 7.9 Hz, 1H), 6.95 (t, J = 7.6 Hz, 1H), 2.23 (s, 3H).
m) 3.1.18. (E)-1-(2-(cyclohexylmethoxy)phenyl)-3-(4-ethylphenyl)prop-2-en-1-one (21-18). Why the enolate signal is missing? Are you sure about the identity of the molecule?
Our response: The correct data is as follows.
1H-NMR (400 MHz, CDCl3): 12.88 (s, 1H), 7.68–7.62 (m, 2H), 7.50 (d, J = 8.1 Hz, 2H), 7.48 (s, 2H), 7.45–7.40 (m, 2H), 7.21 (d, J = 8.1 Hz, 2H), 6.94 (d, J = 8.3 Hz, 1H), 3.83 (d, J = 5.8 Hz, 2H), 2.66 (q, J = 7.6 Hz, 1H), 1.82–1.57 (m, 5H), 1.24 (t, J = 7.6Hz, 3H), 1.19–1.01 (m, 5H).
n) 3.1.19. (E)-3-(4-chlorophenyl)-1-(2-(cyclohexylmethoxy)phenyl)prop-2-en-1-one (21-19). Why the enolate signal is missing? Are you sure about the identity of the molecule?
Our response: The correct data is as follows.
1H-NMR (400 MHz, CDCl3): 12.9 (s, 1H), 7.78–7.74 (m, 4H), 7.69 (d, J = 8.4Hz, 2H), 7.65–7.61 (m, 2H), 7.20 (d, J = 8.4 Hz, 2H), 3.84 (s, 2H), 1.84–1.58 (m, 5H), 1.20–1.02 (m, 5H).
10) The article has been submitted to the Medicinal Chemistry section of Molecules, but a rational discussion about structure-activity relationship is completely missing and should be added in the discussion chapter.
Our response: Based on the structure-activity relationship, benzyloxy, 4-trifluoromomodoxybenzyloxy or cyclohexylmethoxy group is important at the R1 position, and Cl or NHCOCH3 is important at the R3 position. However, H or OH group at the R2 position has little effect on structure-activity relationship. In fact, benzyloxy, 4-trifluoromomodoxybenzyloxy and Cl groups exhibited moderate cytotoxic activity, while cyclohexylmethoxy and NHCOCH3 groups exhibited weak cytotoxic activity. Overall, the anti-melanogenic activity of chalcone derivatives is the strongest when cyclohexylmethoxy and NHCOCH3 groups are located at the R1 and R3 positions. We described these structure-activity relationships with cytotoxicity against anti-melanogenic activity of chalcone derivatives in the discussion section of 2.2.1. Cellular Melanin Production.
11) I’d rather prefer to avoid the use of references in the conclusion paragraph. But this is just a matter of style and I leave to the Editor the final decision about.
Our response: In other some papers published in this journal, references were used in the conclusion paragraph as needed, so we think that using references in the paragraph is not a problem.
Reviewer 2 Report
The presented work if examined without any knowledge of the background of the authors' work, looks very interesting and very well presented. They have used a lot of experimental data to strength their hypothesis about the anti-melanogenic activity. However i got lost when i tried to follow the progress of he work from 2007 to 2016 and finally to the article submitted. On 2007 (https://doi.org/10.1016/j.bmc.2006.10.00) the authors first synthesized most of the derivatives against interleukin-5. On 2016 (https://doi.org/10.1016/j.bbrc.2016.10.110) the same compounds were tested against tyrosinase activity. So the novelty is the elimination of OH in position R2. I would like to have some discussion why the OH derivative from the previous project (2016) didn't show good enough anti-tyrosinase activity. Finally since the crystal structure of Tyrosinase in complex with kojic-acid is available i would to see some insilico data to clarify the optimization of all new compounds. Since the title refers to "optimization" i would like to have more detailed analysis on compound 21-21 compared to derivates lacking the OH group in R2 position. If the introduction, results and discussion section is re-organized based on the history of all derivatives already synthesized by the authors in the past i am positive for the article to be accepted.Author Response
Comments and Suggestions of Reviewer 2
The presented work if examined without any knowledge of the background of the authors' work, looks very interesting and very well presented. They have used a lot of experimental data to strength their hypothesis about the anti-melanogenic activity. However i got lost when i tried to follow the progress of he work from 2007 to 2016 and finally to the article submitted. On 2007 (https://doi.org/10.1016/j.bmc.2006.10.00) the authors first synthesized most of the derivatives against interleukin-5. On 2016 (https://doi.org/10.1016/j.bbrc.2016.10.110) the same compounds were tested against tyrosinase activity. So the novelty is the elimination of OH in position R2. I would like to have some discussion why the OH derivative from the previous project (2016) didn't show good enough anti-tyrosinase activity.
Our response: Thank you for your very insightful comments. Actually, the 26-kinds of chalcone derivatives used in the paper published in 2016 were derived from the paper published at Bioorganic & Medicinal Chemistry in 2007 by Yang et al. In fact, our group was interested in the discovery of anti-melanogenic agents, and at that time, the anti-melanogenic action of the chalcone derivatives was little known. Although the Reviewer made important comments by referring to the paper published in 2016, there was H, benzyloxy or cyclohexylmethoxy group in the R2 position, instead of OH group, and the anti-melanogenic activity was varied. At this time, we fixed four different substituents in R1 and R3 positions and determined the anti-melanogenic activity by OH or H group at R2 position.
Finally since the crystal structure of Tyrosinase in complex with kojic-acid is available i would to see some in silico data to clarify the optimization of all new compounds. Since the title refers to "optimization" i would like to have more detailed analysis on compound 21-21 compared to derivatives lacking the OH group in R2 position. If the introduction, results and discussion section is re-organized based on the history of all derivatives already synthesized by the authors in the past i am positive for the article to be accepted.
Our response: As the Reviewer commented, studies recently have been shown to demonstrate and optimize molecular docking by in silico data for the tyrosinase inhibitory effect of kojic acid or its derivatives. Unfortunately, we didn't prepare the in silico data for the tyrosinase inhibitory effect of compound 21-21, because the anti-melanogenic effect of compound 21-21 is associated with the inhibition of ERK/CREB signaling, but not tyrosinase directly, thereby targeting MITF and MITF-regulated genes including those encoding tyrosinase, TRP1, and TRP2. These results are summarized in the Figure 6. In addition, in the introduction, results and discussion section, we described the detailed analysis on compound 21-21 for anti-melanogenic activity compared to derivatives lacking the OH group in the R2 position and the history of all derivatives already published.
Round 2
Reviewer 1 Report
The request for a major revision was aimed at the correction of some major flaws. These flaws are still present in the paper making it unsuitable for publication.
Reviewer 2 Report
The revised version meets suggestions/corrections and is ready to be accepted by the editor.